# Effects of Topical or Intravitreal Application of Anti-Vascular Endothelial Growth Factor on Density of Intestinal Blood Vessels of Mice

**DOI:** 10.3390/medicina59040809

**Published:** 2023-04-21

**Authors:** Atsushi Ichio, Masahiko Sugimoto, Yuhki Koike, Yuji Toiyama, Mineo Kondo

**Affiliations:** 1Sakuranomori Eye Clinic, Suzuka 510-0226, Japan; atsushi_ichio@yahoo.co.jp; 2Department of Ophthalmology, Mie University Graduate School of Medicine, Tsu 514-8507, Japan; mineo@med.mie-u.ac.jp; 3Department of Gastrointestinal and Pediatric Surgery, Mie University Graduate School of Medicine, Tsu 514-8507, Japan; koikyon@clin.medic.mie-u.ac.jp (Y.K.); ytoi0725@clin.medic.mie-u.ac.jp (Y.T.)

**Keywords:** anti-vascular endothelial growth factor agents, direct titration, intravitreal injection, vasoconstriction

## Abstract

*Background and Objectives:* Anti-vascular endothelial growth factor (anti-VEGF) therapy has become the first-line treatment for diabetic macular edema. However, it is still not clear whether anti-VEGF agents act on systemic blood vessels. The aim of this study is to determine whether a direct topical application or intravitreal injection of anti-VEGF will change the intestinal blood vessels of mice. *Materials and Methods:* C57BL/6 mice were laparotomied under deep anesthesia, and the blood vessels on the surface of the intestines were exposed, examined, and photographed through a dissecting microscope. Vascular changes were evaluated before and at 1, 5, and 15 min after the topical application of 50 µL of the different anti-VEGF agents onto the surface of the intestine (group S) or after the intravitreal injection (group V). The vascular density (VD) was determined for five mice in each group before and after 40 μg/μL of aflibercept (Af), or 25 μg/μL of bevacizumab (Be), or 10 μg/μL of ranibizumab (Ra) were applied. Endothelin-1 (ET1), a potent vasoconstrictor, was used as a positive control, and phosphate-buffered saline (PBS) was used as a control. *Results:* For group S, no significant changes were observed after PBS (baseline, 1, 5, and 15 min: 46.3, 44.5, 44.8, and 43.2%), Be (46.1, 46.7, 46.7, and 46.3%), Ra (44.7, 45.0, 44.7, and 45.6%), and Af (46.5, 46.2, 45.9, and 46.1%, repeated ANOVA) were applied topically. Significant decreases in the VD were observed after ET1 (46.7, 28.1, 32.1, and 34.0%, *p* < 0.05) was topically applied. For group V, no significant differences were observed for all anti-VEGF agents. *Conclusions:* The topical application or intravitreal injections of anti-VEGF agents do not cause a change in the VD of the intestinal vessels, which may be related to its safety.

## 1. Introduction

Anti-vascular endothelial growth factor (anti-VEGF) therapy has become the first-line treatment for diabetic macular edema because the results of many randomized clinical trials have led to successful outcomes [1,2,3]. VEGF was first identified as a major regulator of angiogenesis, and a suppression of VEGF signaling was associated with a decrease in the production of nitric oxide and prostaglandin-I, which relate to vascular constriction. A previous in vivo study also reported that the death of vascular endothelial cells may be the cause of thrombosis [4]. VEGF and related genes have other molecular aspects involved in the progression of proliferative vitreoretinopathy [5].

Additionally, such a suppression can induce vasoconstriction and hypertension, which occurs in approximately 33% of patients receiving systemic anti-VEGF therapy for cancer [6]. In addition, a laser speckle flowgraphy (LSFG) study showed that there was a decrease in the ocular circulation in the treated eye after intravitreal anti-VEGF therapy [7]. Moreover, there are reports that suggested that the plasma VEGF concentration decreased after intravitreal anti-VEGF therapy [8,9,10,11]. From this information, physicians occasionally hesitate to use anti-VEGF therapy because of the risk of systemic vascular complications. A subgroup analysis of the RISE/RIDE and VISTA/VIVID studies showed that frequent anti-VEGF injections at the highest risk of systemic adverse effects, i.e., patients with a recent cerebrovascular accident or myocardial infarction, can induce vascular and mortality events [12]. However, this was the result of frequent applications and may not be applicable in real-world clinical practice. In fact, the safety of the intravitreal anti-VEGF injection has been ensured by the results of its real-world clinical use [13,14]. Thus, a definitive conclusion has not been reached on whether anti-VEGF agents act on systemic blood vessels and cause adverse vascular events. In addition, there are no studies on the direct effects of anti-VEGF agents on the vascular density (VD).

The purpose of this study was to determine the changes in the VD of the intestinal blood vessels of mice from the direct topical application or the intravitreal injection of anti-VEGF agents and evaluate the side effects.

## 2. Materials and Methods

### 2.1. Preparation of Mice for Experiments

Eight- to ten-week-old C57BL/6 mice were purchased from the Japan SLC (Hamamatsu, Shizuoka, Japan). We used 8–10-week-old mice because of the size for observation and minimal effect of aging. The mice were anesthetized with ketamine hydrochloride (100 mg/kg body weight) and xylazine (10 mg/kg body weight). Appropriate deep anesthesia was confirmed by the absence of the withdrawal reflex to a toe pinch, as well as physiological responses, including reduced respiratory and heartbeat rate.

### 2.2. Surgical Procedure for Mice Experiment

The anesthetized mice were placed on a heating pad. A vertical skin incision was made at the lower to middle abdominal area, and the intestines were exposed through an incision of the abdominal muscles. Then, the lateral sides of the abdominal wall were spread apart, which allowed a portion of the intestine to be externalized from the abdominal cavity. One drop of phosphate-buffered saline (PBS) was placed on the intestine to avoid drying.

### 2.3. Preparation and Set Up of Microscope for Experiments

The viewing system was assembled as reported in detail [15]. In brief, the mouse on the heating pad was placed on the stage of a dissecting microscope, and a microscope cover slip attached to an arm of the mouse-holding device was placed on the exposed intestine. The cover slip was placed carefully over the intestine to avoid affecting the gastrointestinal peristalsis (Figure 1). A BX61 Olympus microscope (Tokyo, Japan) equipped with a Hamamatsu Photonics charge-coupled camera (Bridgewater, NJ, USA) was used to take the images.

### 2.4. Administration of Anti-VEGF and Control Agents

Anti-VEGF agents were used with the same concentration as in clinical use: 40 μg/μL of aflibercept (Af; Eylea^TM^, Regeneron Pharmaceuticals, Tarrytown, NY, USA), 25 μg/μL of bevacizumab (Be; Avastin^TM^; Genentech, San Francisco, CA, USA), and 10 μg/μL of ranibizumab (Ra; Lucentis^TM^; Genentech). PBS was used for the control, and 0.1 μg/μL of endothelin-1 (ET1, 4198-s; Peptide Institute, Inc., Osaka, Japan), a potent vasoconstrictor, was used as a positive control [16].

For the experiments, 50 μL of each agent was applied topically into the space between the surface of the intestine and the coverslip (S group). The applied agents spread due to the capillary phenomenon. In the second set of experiments, 1 μL of each agent was injected intravitreally (V group). We performed each experiment for the S and V groups separately. We used only one agent for one mouse for each experiment and a total of five mice was used for each agent. Images were acquired before the application of each agent for both experiments, at 1 min, 5 min, and 15 min after the application (Figure 2a).

### 2.5. Evaluations of Vascular Density

We determined the relative vascular density (VD; %) as the percentage of vascular area to the non-vascular area in the fixed visual field using the ImageJ software (Image processing and analysis in JAVA version 1.53 g; Wayne Rasband, NIH, Bethesda, MD, USA) and AngioTool (free software, downloaded at http://angiotool.nci.nih.gov; (accessed on 1 February 2023). The details of AngioTool were described in our earlier study [17]. Briefly, the images were converted to black–white images using the ImageJ software (Figure 2b). Each black –white image was analyzed using AngioTool and quantitative measurements of the VD were calculated automatically (Figure 2c).

#### Statistical Analyses

All experiments were repeated five times and the means ± standard deviations values are presented. All results were evaluated by 2 masked graders (AI and MS) and the average value was used for the statistical analyses. Data were analyzed by two-way repeated analysis of variance (ANOVA) followed by Bonferroni post hoc tests for the comparison of the means. Statistical significance was set at a *p* < 0.05.

## 3. Results

### 3.1. Vascular Density (VD) after Applications of Control Agents

The PBS and ET1 were used as controls, and raw images were shown in Figure 3. No obvious vascular changes were observed for the PBS-treated groups after direct application (S) or intravitreal injections (V group). For the ET1-treated group, there was a significant vasoconstriction from 1 to 15 min. No obvious changes were observed in the V group for all agents and controls. The mean VD of the PBS-treated S group was 46.3 ± 1.1% at the baseline, 44.5 ± 0.8% at 1 min, 44.8 ± 0.4% at 5 min, and 43.2 ± 1.4% at 15 min. None of the changes was significant (*p* = 0.40, repeated ANOVA; Table 1). The mean VD of the PBS-treated V group was 44.8 ± 0.4% at the baseline, 45.2 ± 0.6% at 1 min, 45.4 ± 0.3% at 5 min, and 45.7 ± 0.1% at 15 min. None of the changes was significant (*p* = 0.86).

There were significant decreases in the VD in the ET1-treated S group at 1 and 5 min after the applications. The reduction in the relative VD after ET1 indicates that the constricted vessels were not detected by the image analyzer. The mean VD of the ET1-treated S group was 46.7 ± 5.7% at the baseline, 28.1 ± 8.6% at 1 min (*p* = 0.008), 32.1 ± 15.2% at 5 min (*p* = 0.001), and 34.0 ± 3.8% at 15 min.

The mean VD of the ET1-treated V group was 45.4 ± 5.7% at the baseline, 43.5 ± 4.2% at 1 min, 38.7 ± 8.2% at 5 min, and 40.7 ± 6.7% at 15 min. None of the changes was significant (*p* = 0.58).

### 3.2. Evaluations of Vascular Density for Different Anti-VEGF Agents

Raw images of anti-VEGF-treated groups are shown in Figure 4. No obvious vascular changes were observed for the S and V groups after each anti-VEGF treatment. The mean VD of the bevacizumab-treated S group was 46.1 ± 0.4% at the baseline, 46.7 ± 0.4% at 1 min, 46.7 ± 0.4% at 5 min, and 46.3 ± 0.6% at 15 min. None of the changes was significant (Table 2, *p* = 0.70). The mean VD of the ranibizumab-treated S group was 44.7 ± 1.3% at the baseline, 45.0 ± 2.1% at 1 min, 44.7 ± 2.1% at 5 min, and 44.6 ± 2.6% at 15 min. None of the changes was significant (*p* = 0.95). The mean VD of the aflibercept-treated S group was 46.5 ± 1.2% at the baseline, 46.2 ± 0.9% at 1 min, 45.9 ± 1.1% at 5 min, and 46.1 ± 1.3% at 15 min. None of the changes was significant (*p* = 0.70).

The mean VD of the bevacizumab-treated V group was 45.0 ± 0.5% at the baseline, 45.4 ± 0.6% at 1 min, 44.7 ± 0.6% at 5 min, and 45.6 ± 0.4% at 15 min. None of the changes was significant (*p* = 0.14). The mean VD of the ranibizumab-treated V group was 45.8 ± 0.5% at the baseline, 46.2 ± 0.4% at 1 min, 45.2 ± 1.3% at 5 min, and 45.6 ± 0.3% at 15 min. None of the changes was significant (*p* = 0.26). The mean VD of the aflibercept-treated V group was 47.0 ± 0.2% at the baseline, 46.9 ± 0.1% at 1 min, 46.6 ± 0.7% at 5 min, and 46.9 ± 0.1% at 15 min. None of the changes was significant (*p* = 0.63).

## 4. Discussion

Our results showed that none of the three anti-VEGF agents affected the relative VD of mouse intestinal vessels after direct application or after an intravitreal injection. Additionally, the lack of a change with all agents, including ET1, after intravitreal injection—though topical ET1 administration causes vascular constriction—suggests that they did not reach to the intestinal vessels in high enough concentrations. The results of randomized clinical trials have shown that anti-VEGF therapy for DME did not enhance the systemic vascular disease [18,19], and its safety has been affirmed in real-world clinical practice [13,14]. In addition, an inhibition of VEGF can induce vascular changes by altering a group of molecules downstream of VEGF, resulting in vasoconstriction and other changes. We have reported no significant changes in the factors involved in vascular infarction after the intravitreal injection of aflibercept and ranibizumab [20]. These findings of safety support our results of the present study: a lack of changes in the intestinal vessels.

In this study, we quantified the changes relatively simply in the VD after topical applications with our technique. It is known that VEGF receptors are expressed in the endothelial cells of the small intestine [21]. In addition, there are reports of gastrointestinal perforations after systemic bevacizumab administration, suggesting a possibility of intestinal necrosis associated with the systemic application of anti-VEGF agents [22]. We evaluated the intestinal vessels of mice, and we considered that this model was appropriate for evaluating the effects of anti-VEGF agents. However, cerebrovascular and cardiovascular complications have been reported following intravitreal anti-VEGF injections. Although earlier reports stated that long-term and frequent use of anti-VEGF agents may cause cerebrovascular complications and death, the studies did not consider other vascular injuries including the heart [12]. This is because different blood vessels in the different organs may tolerate different doses and agents. A study has shown that after an intravenous injection of anti-VEGF-A, it accumulates in the heart, pancreas, and bladder [23]. This implies that the anti-VEGF agents do not spread to the different organs equally. It does suggest that different blood vessels may respond differently to the anti-VEGF agents, and such vascular injury may be organ specific. We need further examinations either using cerebral or cardiovascular vessels.

We compared three humanized anti-VEGF antibodies, viz., aflibercept, bevacizumab, and ranibizumab, on the intestines of mice. There is some debate on whether they act on the blood vessels of mice because these agents were developed to be human specific. Earlier studies reported the effect was weaker than mouse anti-VEGF-A antibody and ranibizumab [24,25], and bevacizumab bound to human VEGF-A at a 5-log lower concentration than either mouse or rat VEGF-A [26]. However, the results of two earlier studies reported that bevacizumab was also active in mice [27,28]. It is also not clear whether aflibercept is functional in mice. Although we did not observe any vasoconstriction after the administration of anti-VEGF agents on the mice intestines, it is not clear whether a higher concentration of the agents may affect the vessels. In addition, we did not select any one of the VD-modulating agents. On these points, we need further studies to resolve these questions.

There are some other limitations to our study. First, we evaluated healthy vessels. Long-term hyperglycemia in patients with diabetes may result in vascular damage compared to healthy individuals, and the vessels’ ability to tolerate the agents may be different. Therefore, it is important to evaluate abnormal vessels using an acute hyperglycemia model with streptozotocin administration or diabetic model mice, such as the Akita or db/db mice. Second, the S group evaluated the effects of extravascular penetration, not the intravenous effect. We also need an intravenous administration model. In addition, there is a possibility that the lack of a vascular change after intravitreal injection may be due to the fact that they did not penetrate the intestinal vessels with sufficiently high concentrations to induce vascular change. Third, although we evaluated vascular changes using the VD as a parameter, we did not distinguish between arterial and venous vessels. Because there are anatomic differences between arterial and venous vessels, an analysis should be performed separately between the two types of vessels. Fourth, we used the method of acute effect of anti-VEGF agents within 15 min. We need an additional experiment to evaluate the chronic effect of anti-VEGF agents. Finally, our observations were made at low magnification. A more detailed examination of the vascular architecture must be made using other devices such as two-photon laser-scanning microscopy [29]. Although we evaluated the changes by two-photon laser-scanning microscopy (Appendix A), it was difficult to image the same site during the 15 min of observation due to gastrointestinal peristalsis. Improvement of the observation systems is needed in the future.

## 5. Conclusions

In this study, anti-VEGF agents did not cause the VD changes within the short term. Neither topical application nor intravitreal injections of three anti-VEGF agents caused a vasoconstriction of the intestinal vessels of healthy mice. Although this may imply the safety of anti-VEGF agents, evaluations of pathological vessels using diabetic models using advanced instruments are necessary.

## Figures and Tables

**Figure 1 medicina-59-00809-f001:**
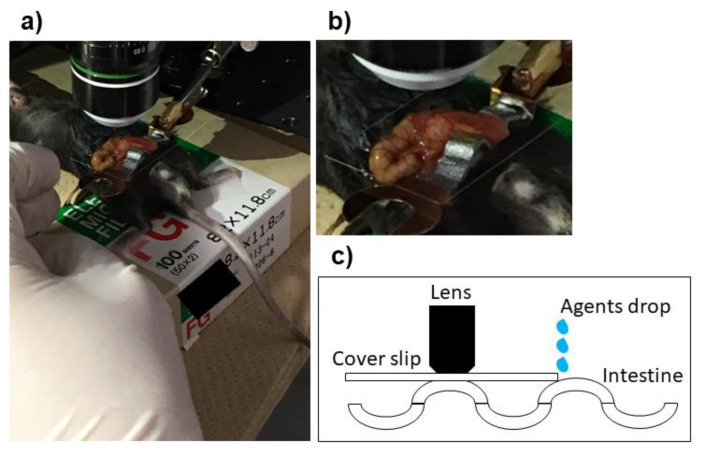
Preparation of mouse and microscope set up. A vertical skin incision is made on a deeply anesthetized C57BL/6 mouse. The intestines are exposed, and the mouse was placed on the stage of a dissecting microscope. A microscope cover glass was attached to the end of an arm of the mouse-holding device and adjusted to avoid excess pressure on the intestines (**a**,**b**). Fifty microliters of each agent were applied into the space between the surface of the intestine and the coverslip for the S group (**c**).

**Figure 2 medicina-59-00809-f002:**
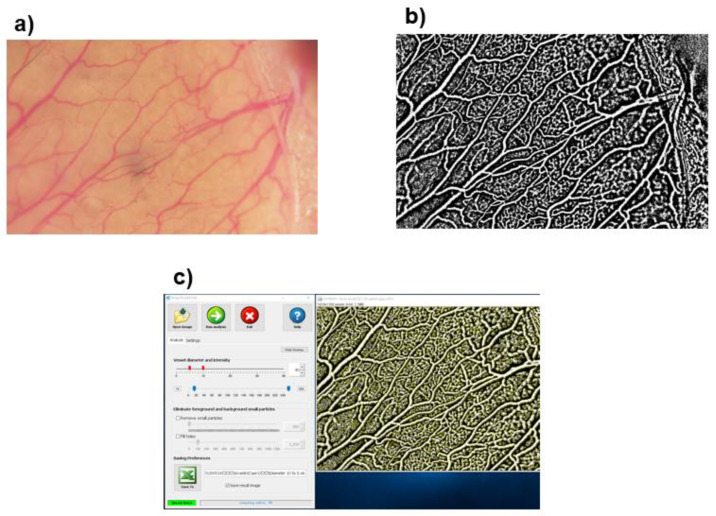
Evaluations of vascular density. The recorded images (**a**) were converted to black–white images using ImageJ (**b**). The blood vessels were displayed in black, and the surrounding tissue in white. Each black–white image was analyzed using AngioTool and the vascular density was calculated automatically (**c**).

**Figure 3 medicina-59-00809-f003:**
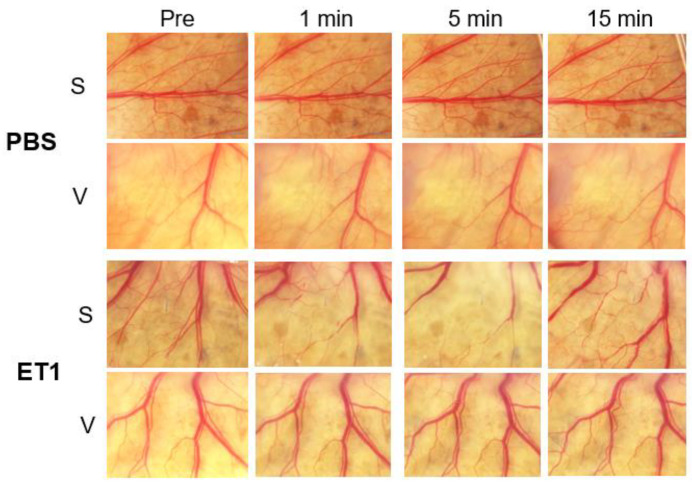
Vascular density evaluations for control agents. No obvious vascular changes were observed for PBS-applied blood vessels of the S and V groups. For the ET1-treated group, obvious vasoconstrictions were observed for the S group (second panel from the bottom) from 1 to 15 min. No obvious changes were observed for the V group (bottom panel). ET-1, endothelin-1; PBS, phosphate-buffered saline.

**Figure 4 medicina-59-00809-f004:**
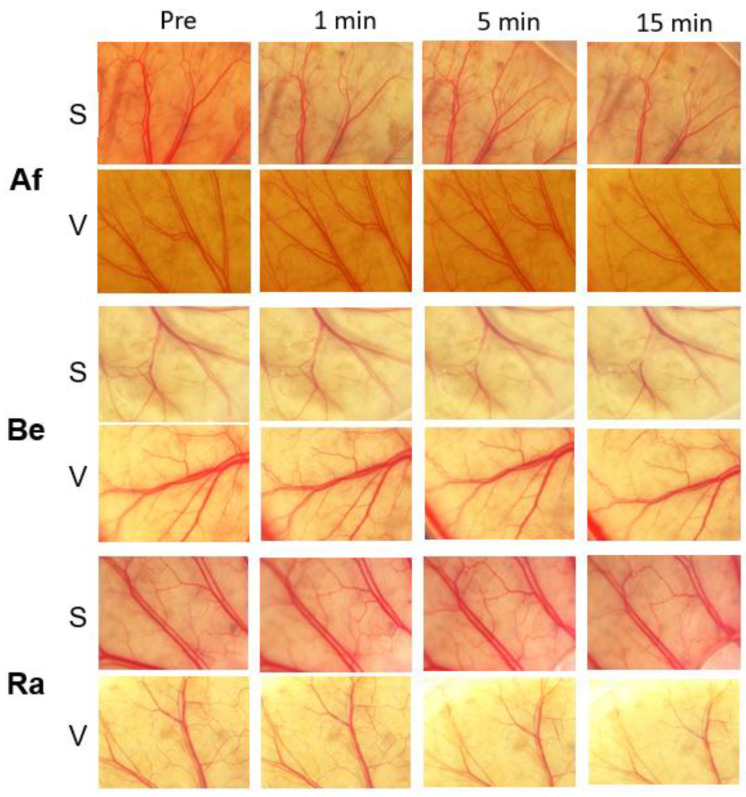
Vascular density evaluations for different anti-VEGF agents. No obvious vascular change was observed for each anti-VEGF agent in the treated group (both S and V groups). VEGF: vascular endothelial growth factor.

**Table 1 medicina-59-00809-t001:** Vascular density changes for control agents.

		Pre (%)	1 min (%)	5 min (%)	15 min (%)
PBS	S	46.3 ± 1.1	44.5 ± 0.8	44.8 ± 0.4	43.2 ± 1.4
	V	44.8 ± 0.4	45.2 ± 0.6	45.4 ± 0.3	45.7 ± 0.1
ET1	S	46.7 ± 5.7	28.1 ± 8.6 **	32.1 ± 15.2 *	34.0 ± 3.8
	V	45.4 ± 5.7	43.5 ± 4.2	38.7 ± 8.2	40.7 ± 6.7

(*n* = 5, repeated ANOVA, *: *p* < 0.05, **: *p* < 0.01). S, surface of intestine drop group; ET1, endothelin-1; V, intravitreous injection group; PBS, phosphate-buffered saline.

**Table 2 medicina-59-00809-t002:** Vascular density changes for three anti-VEGF agents.

		Pre (%)	1 min (%)	5 min (%)	15 min (%)
Bevacizumab	S	46.1 ± 0.4	46.7 ± 0.4	46.7 ± 0.4	46.3 ± 0.6
	V	45.0 ± 0.5	45.4 ± 0.6	44.7 ± 0.6	45.6 ± 0.4
Ranibizumab	S	44.7 ± 1.3	45.0 ± 2.1	44.7 ± 2.1	44.6 ± 2.6
	V	45.8 ± 0.5	46.2 ± 0.4	45.2 ± 1.3	45.6 ± 0.3
Aflibercept	S	46.5 ± 1.2	46.2 ± 0.9	45.9 ± 1.1	46.1 ± 1.3
	V	47.0 ± 0.2	46.9 ± 0.1	46.6 ± 0.7	46.9 ± 0.1

(*n* = 5, repeated ANOVA). S, surface of intestine drop group; V, intravitreous injection; VEGF, vascular endothelial factor.

## Data Availability

The datasets used during the current study are available from the corresponding author on request.

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
