# Peer review of "Effects of Topical or Intravitreal Application of Anti-Vascular Endothelial Growth Factor on Density of Intestinal Blood Vessels of Mice"

_medicina, 2023, doi:10.3390/medicina59040809_

Round 1

Reviewer 1 Report

Dear Dr.,

Title: Effects of topical or intravitreal application of anti-vascular endothelial growth factor on density of intestinal blood vessels of mice

Manuscript ID: medicina-2314037

Overall comments: Authors described in this manuscript: The effect of anti-vascular endothelial growth factor agents (Aflibercept, bevacizumab and ranibizumab) on intestinal vascular density level in mice by two different routes (topical or intravitreal) of administration. The limitation of this study author focused only in intestinal vascular density changes whereas if retinal vascular density is also made will be more strengthen in this manuscript. The overall manuscript is written well and it has novelty in this field of research.

Specific comments:

1.      Authors used the method as acute effect of anti-VEGF agents. If chronic effect of anti-VEGF agents need to evaluate to proven these research output.

2.      A microscope cover glass was placed on the intestinal tissue. Thereafter drugs were applied, how author ensured the drug was applied in intestinal area where coverslip was placed.

3.      Methodology can be making with more sections like animal used, surgical procedure, and drug administration for better understanding.

Minor comments

1.      In the abstract, the background of the study statement can add at the beginning of abstract.

2.      How long time given for the assessment of vascular density between intravitreal injection and intestinal vascular density assessment.

3.      What is the reason author selected Eight- to ten-week-old C57BL/6 mice in this study.

4.      If author selected any one of the vascular density modulating agents may give additional weightage for the demonstration of anti-VEGF effects.

5.      Figures 3 and 4 legends, must be mentioned the abbreviation of term used in figures.

6.      Conclusion section first statement need to recheck, anti-VEGF agents cause the VD changes or not.

7.      Most of the references are too old.

*****

Author Response

Reply to reviewer 1

Specific comments:

  1. Authors used the method as acute effect of anti-VEGF agents. If chronic effect of anti-VEGF agents need to evaluate to proven these research output.
  2. A) We add explanation about this point (L289).

  1. A microscope cover glass was placed on the intestinal tissue. Thereafter drugs were applied, how author ensured the drug was applied in intestinal area where coverslip was placed.
  2. A) We add explanation about this point (L93).

  1. Methodology can be making with more sections like animal used, surgical procedure, and drug administration for better understanding.
  2. A) We add more detailed sections in “Method”.

Minor comments

  1. In the abstract, the background of the study statement can add at the beginning of abstract. A)We add explanation in the abstract (L12).
  2. How long time given for the assessment of vascular density between intravitreal injection and intestinal vascular density assessment. A) We did not performed experiment for S-group and V-group for same mice. We add explanation about this point (L95).
  3. What is the reason author selected Eight- to ten-week-old C57BL/6 mice in this study. A) We add explanation about this point (L65).
  4. If author selected any one of the vascular density modulating agents may give additional weightage for the demonstration of anti-VEGF effects. A) We add explanation about this point (L274).
  5. Figures 3 and 4 legends, must be mentioned the abbreviation of term used in figures. A) We add explanation about the abbreviation of terms for Figure 3 and 4.
  6. Conclusion section first statement need to recheck, anti-VEGF agents cause the VD changes or not. A) We add explanation about this point (L298).
  7. Most of the references are too old. A) We delete some old references (Ref4, 23).

Reviewer 2 Report

Authors wrote an interesting article

Improvements: please expand introduction

Have a look and cite to this paper in introduction, PMID: 24227910 about VEGF and genes

please improve the limitations of the study

Author Response

Reply to reviewer 2

  1. Improvements: please expand introduction. Have a look and cite to this paper in introduction, PMID: 24227910 about VEGF and genes A) We add this article as reference and expand introduction (L40).
  2. please improve the limitations of the study A) We add explanation about this point (L289).

Round 2

Reviewer 2 Report

Accept